# Total Hysterectomy by Low-Impact Laparoscopy to Decrease Opioids Consumption: A Prospective Cohort Study

**DOI:** 10.3390/jcm11082165

**Published:** 2022-04-13

**Authors:** Yohann Dabi, Samia Ouasti, Hélène Didelot, Henri Wohrer, Dounia Skalli, Gregoire Miailhe, Jennifer Uzan, Clément Ferrier, Sofiane Bendifallah, Bassam Haddad, Emile Daraï, Cyril Touboul

**Affiliations:** 1Sorbonne University, 75006 Paris, France; sofiane.bendifallah@aphp.fr (S.B.); emile.darai@aphp.fr (E.D.); cyril.touboul@aphp.fr (C.T.); 2Obstetrics and Gynecology Department, Tenon Hospital, Assistance Publique des Hôpitaux de Paris (AP-HP), 75020 Paris, France; samia.ouasti@aphp.fr (S.O.); clement.ferrier@aphp.fr (C.F.); 3Obstetrics and Gynecology Department, Intercommunal Hospital of Creteil, 94000 Paris, France; helene.didelot@aphp.fr (H.D.); henri.worhrer@aphp.fr (H.W.); dounia.skalli@chicreteil.fr (D.S.); gregoire.miailhe@chicreteil.fr (G.M.); jennifer.uzan@chicreteil.fr (J.U.); bassam.haddad@chicreteil.fr (B.H.); 4University of Medicine Paris Est Créteil, 94000 Créteil, France

**Keywords:** low impact laparoscopy, hysterectomy, postoperative pain, opioids

## Abstract

Our objective was to evaluate postoperative pain and opioid consumption in patients undergoing hysterectomy by low-impact laparoscopy and compare these parameters with conventional laparoscopy. We conducted a prospective study in two French gynecological surgery departments from May 2017 to January 2018. The primary endpoint was the intensity of postoperative pain evaluated by a validated numeric rating scale (NRS) and opioid consumption in the postoperative recovery unit on Day 0 and Day 1. Thirty-two patients underwent low-impact laparoscopy and 77 had conventional laparoscopy. Most of the patients (90.6%) who underwent low-impact laparoscopy were managed as outpatients. There was a significantly higher consumption of strong opioids in the conventional compared to the low-impact group on both Day 0 and Day 1: 26.0% and 36.4% vs. 3.1% and 12.5%, respectively (*p* = 0.02 and *p* < 0.01). Over two-thirds of the patients in the low-impact group did not require opioids postoperatively. Two factors were predictive of lower postoperative opioid consumption: low-impact laparoscopy (OR 1.38, 95%CI 1.13–1.69, *p* = 0.002) and a mean intraoperative peritoneum below 10 mmHg (OR 1.25, 95%CI 1.03–1.51). Total hysterectomy by low-impact laparoscopy is feasible in an outpatient setting and is associated with a marked decrease in opioid consumption compared to conventional laparoscopy.

## 1. Introduction

Around 33,000 hysterectomies are performed each year in the UK, which means that one in five women will undergo this major gynecological surgery at some point [1]. In the past, hysterectomies were performed by the vaginal route or by open surgery (laparotomy), but the development of laparoscopy has supplanted these approaches over the years in developed countries [2]. The emergence of robot-assisted surgery, which reduces the surgical complexity, has further promoted the laparoscopic approach allowing more gynecologists to perform the procedure [3]. These combined improvements have led to reduced hospital stays and even to outpatient management in certain cases. Overall, there is a trend for increased outpatient management for surgical procedures as it is associated with greater patient satisfaction and lower costs compared to inpatient management [4,5,6].

Nevertheless, surgeons need to be sure that their patients will not experience any pain once home. Pain typically leads to the over-prescription and over-consumption of opioids. In a cohort of 40,000 patients who had undergone surgery, Clarke et al. reported that 49% were discharged with an opioid prescription, and that 3% were still using opioids 90 days after the operation [7]. The opioid crisis that has come to light in the United States over the past few years has largely been driven by postoperative prescriptions for pain management. In a study by Brummet et al. of a large cohort of privately insured patients in the United States, the rates of new persistent postoperative opioid use ranged from 5.9% to 6.5% [8], even after minor surgery. The rates of new persistent opioid use were close to 6% after hysterectomy in their study. Postoperative opioid use is thus a current public health crisis, and surgeons can play their part by improving surgical procedures. 

Laparoscopic surgery has evolved significantly since the first laparoscopic hysterectomy by Reich in 1989 [9]. Mini-invasive surgery now encompasses techniques such as robot-assisted surgery, single-port laparoscopy, and, more recently, the concept of “low-impact laparoscopy”. Low-impact laparoscopy combines low peritoneal pressure with the miniaturization of laparoscopic ports from 5–15 mm to 3–5 mm in diameter. Although this technique has been shown to reduce postoperative pain and increase esthetic satisfaction for several surgical procedures [10,11], there are very few data concerning gynecologic surgery. In a cohort of 60 patients, Sroussi et al. reported the feasibility and effectiveness of low-impact laparoscopy for reducing shoulder pain after benign gynecological surgeries [12]. To date, little attention has been paid to postoperative analgesic and opioid consumption in patients undergoing a hysterectomy by low-impact laparoscopy.

The main objective of our study was thus to evaluate postoperative pain and opioid consumption in patients undergoing a hysterectomy by low-impact laparoscopy. Our second objective was to compare these parameters with those of patients undergoing a conventional laparoscopic hysterectomy.

## 2. Methods

### 2.1. Population

We conducted an analysis of patients prospectively included, from May 2017 to January 2018, in two French surgical gynecology departments (Centre Hospitalier Intercommunal de Creteil and Tenon Hospital, AP-HP). All consecutive adult patients undergoing laparoscopic hysterectomy were included in the study. Patients with additional procedures were excluded from the analysis. The research protocol was approved by the Institutional Review Board (IRB) of the French College of Obstetrics and Gynecology (CEROG 2021—GYN—0101).

### 2.2. Surgical Details

All the low-impact laparoscopic procedures were performed by two senior surgeons. Conventional laparoscopy was performed by other surgeons in the departments.

Patients in the low-impact group were all prescribed analgesics including paracetamol, a non-steroidal anti-inflammatory drug (NSAID), and a weak opioid (tramadol) at their preoperative appointment. All the low-impact laparoscopy hysterectomies were planned as outpatient procedures.

### 2.3. Anesthesia Protocol

A formalized anesthesia protocol was applied for patients who underwent low-impact laparoscopy. Anesthesia was achieved by a remifentanil/propofol rapid induction technique, and curarization by rocuronium bromide (quick-acting). All the patients received an infusion of ketamine (0.15 mg/kg) at the beginning of the surgery and were administered an antiemetic (dexamethasone) during the procedure. Peroperative analgesia also included paracetamol and nefopam. Vascular filling was restricted to 2 to 3 mL/kg/h.

A bladder catheter was inserted systematically for both procedures.

For the low-impact procedures, pneumoperitoneum was created at the umbilicus using a Verress needle up to 15 mmHg. An AirSeal^®^ 5 mm valve-free trocar—which provides stable pneumoperitoneum even under constant suction—was then introduced [13,14]. We then inserted a 3 or 4 mm 30° optical lens. Under visual control, three other ports were introduced: in the right and left iliac fossae, and in the right hypochondrium. The optical lens was then moved to the right hypochondrium port for the rest of the surgery, and intraperitoneal pressure was lowered to 6–8 mmHg.

For the conventional laparoscopic procedure, a 10 mm optical lens was introduced through the umbilicus. The first insufflation was at 15 mmHg. It was then mediated either by a conventional insufflator or by an Airseal^®^ device. Three other 5 mm ports were introduced under direct visual control in the right and left iliac fossae and medially. The pressure was then lowered to 10–12 mmHg.

The operative technique was similar in the two groups. Hysterectomy was performed using Ultracision^®^ Harmonic forceps and coagulation using a bipolar grasp. At the end of surgery, the ports were infiltrated by ropivacaine (75 mg) at the surgeon’s discretion.

Postoperative management was similar in the two groups, especially in terms of pain evaluation and administration of analgesics. The nurses in the postoperative recovery unit (PACU) and outpatient/gynecologic unit were completely blinded to the surgical approach used.

The bladder catheter was removed at the end of surgery for the outpatients and on Day 1 for inpatients. 

### 2.4. Outcome Measures

The primary endpoint was opioid consumption for postoperative pain. Pain was evaluated in the PACU on Day 0 and Day 1 by a validated numeric rating scale (NRS) ranging from 0 to 10 (where 0 is no pain and 10 the worst imaginable pain) [15,16,17,18]. Opioid consumption was recorded at each time point. Weak opioids included tramadol 50 mg 4 times a day, and strong opioids consisted of morphine titration by 1 mg doses.

Appropriate analgesics were administered according to the NRS. Postoperative analgesic prescriptions included paracetamol, nefopam, NSAIDs, and tramadol to be used only if the other drugs were not sufficient for pain relief. According to the NRS: paracetamol was used for light pain (NRS < 4), ibuprofen or weak opioids (Tramadol) were used for moderate pain (4 ≤ NRS < 7), and morphine was used for intense pain (NRS ≥7).

The outpatients were all called on Day 1 as standard protocol to ensure the absence of uncontrolled pain.

### 2.5. Statistical Analysis

Databases were managed using Excel (Microsoft Corporation, Redmond, WA, USA), and statistical analyses were performed using R studio software (1.1.463 version, available online).

After descriptive analysis, the patients in the low-impact laparoscopy group were compared to those in the conventional laparoscopy group. Data are presented as *n*(%) unless otherwise specified. Statistical analysis was based on Student’s t test for continuous variables and the χ^2^ test or Fisher’s exact test for categorical variables. All the factors that could influence postoperative pain and analgesic consumption were then investigated in univariate analysis. Values of *p* < 0.05 were considered to denote significant differences.

## 3. Results

### 3.1. Clinical Characteristics and Surgical Parameters

During the study period, of the 109 patients who had a hysterectomy by laparoscopy, 32 underwent a low-impact laparoscopic procedure and 77 a conventional laparoscopy. The data are shown in Table 1. The main clinical characteristics were comparable between the two groups with no significant difference for age, obesity, American Society of Anesthetists (ASA) physical status score, comorbidities, or history of surgery (including cesarean sections) (*p* > 0.05). 

Surgical variables are displayed in Table 2. The indication for hysterectomy was more likely to be of an oncologic nature for the patients in the low-impact group (43.8% (14/32) versus 23.4% (14/77), *p* = 0.01). The size of the uterus was comparable in the two groups with 63 (21.3) mm and 65 (22.5) mm and in the low-impact and conventional groups, respectively (*p* = 0.61).

Patients in the low-impact group had more complementary surgery such as ureterolysis (15.6% (5/32) versus 1.3% (1/77)), or omentectomy/appendicectomy (*p* < 0.001).

There was no significant difference in the duration of surgery, but the duration of anesthesia was greater in patients the conventional procedure group (193 versus 174 min, *p* = 0.03). None of the patients of the total cohort experienced significant perioperative bleeding.

### 3.2. Postoperative Pain and Opioid Consumption

The main postoperative outcomes are displayed in Table 3. Most of the patients in the low-impact group were managed as outpatients (90.6%, 29/32). In the whole cohort, two patients (one in each group) experienced operative complications that required a second procedure: one patient in the low-impact group experienced a late vesicovaginal fistula, which was successfully repaired 3 months later, and the other (in the conventional group) a hemorrhage on the umbilical port. 

The maximum pain evaluated in the PACU was significantly higher in the patients that had conventional laparoscopy (*p* = 0.04) but similar on Days 0 and 1 postoperatively (*p* = 0.92 and *p* = 0.32, respectively). NSAIDs were used for two patients (2.6%) in the conventional group and for one (3.1%) in the low-impact group. 

No patient (0.0%) consumed weak opioids in the PACU or on Day 0 in the low-impact group.

There was a significantly higher consumption of strong opioids in the conventional group vs. the low-impact group on Day 0 and Day 1: 26.0% (20/77) and 36.4% (28/77) vs. 3.1% (1/32) and 12.5% (4/32), respectively (*p* = 0.02 and *p* <0.01). Of note, more than two-thirds of the patients who underwent a low-impact procedure did not use any opioids in the postoperative period. Postoperative opioid consumption and pain evaluation are plotted in Figure 1.

### 3.3. Factors Associated with Postoperative Opioid Consumption

Two factors were predictive of lower postoperative opioid consumption: a surgical approach by low-impact laparoscopy (OR 1.38 95%CI 1.13–1.69), *p* = 0.002) and a mean intraoperative peritoneum pressure below 10 mmHg (OR 1.25 95%CI 1.03–1.51). None of the other parameters, including uterus size and duration of pneumoperitoneum, had an influence of postoperative opioid consumption, (Table 4).

## 4. Discussion

### 4.1. Main Findings

We report here that patients who underwent a hysterectomy by low-impact laparoscopy experienced less postoperative pain and used significantly fewer opioids than those who had conventional laparoscopy. More than two-thirds of the patients who underwent a low-impact procedure did not use any opioids postoperatively. Low-impact laparoscopy was successfully managed in an outpatient setting for around 90% of the patients. Two factors were predictive of lower postoperative opioid consumption: low-impact laparoscopy, and a mean intraoperative peritoneum pressure below 10 mmHg.

### 4.2. Interpretation

The implementation of low-impact procedures in our centers was motivated by the double objective of achieving outpatient management and reducing (and even completely suppressing) postoperative opioid consumption.

It is now accepted that hysterectomy by laparoscopy is associated with lower morbidity and a shorter hospital stay than hysterectomy by laparotomy or by the vaginal route, for both cancerous and benign indications [19,20]. For many years, the challenge has been to improve surgical skills to decrease the number of cases requiring a laparotomy. During the last decade, the exponential use of robotic-assisted surgery has democratized minimally invasive surgery to less experienced surgeons [1,21]. Several techniques have been developed with a view to reducing surgical morbidity, including a reduction in the number of ports as described by Tyan et al. [22]. The miniaturization of laparoscopic instruments—an approach known as mini-laparoscopy—represents little difficulty for trained surgeons, as the lower grasping ability is rapidly overcome. In our setting, we chose to use a 3 mm port in the upper-right region, which enables the use of Ultracision^®^ Harmonic forceps energy in the 5 mm umbilical port or even larger bipolar forceps in the case of uncontrolled hemorrhage or large uterine pedicles. Mini-laparoscopy has been used by general surgeons for various procedures, such as cholecystectomy, with good outcomes. Casarin J. et al. in a recent review concluded that mini-laparoscopy could be a valid alternative to conventional laparoscopy in gynecologic surgery including major surgical and oncological procedures [23]. In our cohort, the two groups (low impact and conventional laparoscopy) did not differ for the main clinical characteristics, with the same proportion of older/obese patients at higher risk of complex procedures and postoperative complications. We observed one case of vesico-vaginal fistula in the low-impact group caused by a vaginal vault stich. This rare complication could have been avoided by a more distal vesico-vaginal cleavage and was certainly independent of the approach.

A recent report from the West Sussex NHS Trust highlighted that the UK is now heading toward a US, opioid-style crisis with five people dying every day from opioid overdose [24]. There is now abundant literature on this matter reporting that a significant proportion of opioid naïve patients prolong postoperative consumption [25,26,27]. The responsibility of postoperative pain management is shared between surgeons and anesthesiologists. Although it is true that not all surgeries are equally painful and therefore do not require the same amount of analgesics, little is known about the quantity each procedure could require. One major issue is the lack of patient reporting of opioid consumption and the lack of specific prescription guidelines either for the recovery room or on hospital discharge. A recent systematic review by Brenton et al. concluded that there was an absence of evidence for opioid-free hysterectomy [28]. In our cohort, five patients (15.6%) used opioids on the day of surgery (Day 0) or the day after (Day 1), which was significantly lower than in the conventional laparoscopy group (51 patients, 66%; *p* < 0.001). The use of opioids did not differ by surgical indication (oncological or benign). The use of non-opioid medication was also significantly lower in the low-impact group. Several authors have studied the performance of hysterectomy by mini-laparoscopy but with a focus on feasibility and complications as well as cosmetic results [3,29,30]. Although the miniaturization of instruments from 5 to 3 mm does indeed provide a cosmetic advantage, we feel that the main benefit of mini-laparoscopy lies in the clinically relevant reduction in postoperative pain. Our low-impact approach associates mini-laparoscopy with low pressure (<10 mmHg). Intraperitoneal pressure is commonly set between 12 and 15 mmHg [31], and sometimes higher. Although guidelines recommend that “the lowest pressure allowing adequate exposure of the operative field” is to be used [32], our work clearly demonstrates that lower pressures are feasible and indeed beneficial, with comparable surgical duration, pneumoperitoneum duration, and no increased rates of per- or postoperative complications. Low pressure, which is usually defined as a pressure ranging from 6 to 10 mmHg [33,34,35,36], potentializes the effect of instrument miniaturization.

D’Angelis et al. reported that low-impact laparoscopy in the setting of cholecystectomy provided improved postoperative outcomes and less pain [10]. Some authors argue that mini-laparoscopy is only a valid technique for highly selected patients [23]. In our cohort, 32 patients benefited from the low-impact procedure, and they were comparable with those elected for conventional surgery. This implies that the limits of low-impact or mini-procedures might be the same as those of laparoscopy in general: comorbidities and surgical history, very large uterus, etc.

### 4.3. Limitations

Our study has some limitations. Two experienced surgeons performed all the low-impact procedures, which might have resulted in an overestimation of the benefit of the technique, since surgical duration and gestures performed are based on a surgeon’s skills. However, in their experience, the main barriers for undertaking a low-impact procedure were concerns about lacking adequate visibility with low pressure and 3 mm optical lens, and concerns about mastering the mini-instruments. It deserves to be mentioned that the transition in our centers from conventional to micro-instruments was responsible for some material difficulties, and the operative room staff were unfamiliar with the new instruments for the first few cases. These data are not registered in this work. Another limitation is the short follow-up for postoperative pain. We only evaluated pain on Day 0 and Day 1 postoperatively, and some patients might experience prolonged or delayed pain. However, acute pain requiring opioids is usually the most intense on the first two days following surgery. Another limitation of our work was the inability to evaluate the impact of peroperative bleeding on postoperative pain, since none of the patients experienced significant bleeding. This could be explained by two factors: (1) the experience of the surgeons that performed the surgeries or (2) the relatively limited sample size. Another limitation was the lack of data regarding anesthetic management especially pulmonary recruitment maneuver that could significantly reduce the need for postoperative opioids. These data were not part of our data collection initially and should be included in the future RCT that will be conducted. Finally, bladder catheterization ablation on postoperative day 1 versus immediately at the end of surgery in the low-impact group may be a confounding factor, as it is a source of discomfort for patients.

The most appropriate method to formally assess the benefit of low impact procedures would be to conduct a randomized controlled trial. In Figure 2, we present the theoretical number of subjects that would be necessary to include in order to demonstrate the benefit of the low-impact procedure on the reduction in ever consuming opioids in the postoperative period. In the setting of 60% of patients consuming opioids in the conventional group, showing a 50% reduction in opioid consumption would require the inclusion of 112 patients. 

## 5. Conclusions

Our study shows that, for women undergoing total hysterectomy, low-impact laparoscopy and an intraperitoneal pressure of <10 mmHg are the main determinants of reduced postoperative pain and analgesic consumption. Total hysterectomy by low-impact laparoscopy is feasible and can be performed in an outpatient setting. Low-impact procedures could constitute the best option for reducing (and possibly eliminating) opioid consumption in the postoperative period. A randomized controlled trial should now be performed to definitely assess the potential benefit of low-impact procedures.

## Figures and Tables

**Figure 1 jcm-11-02165-f001:**
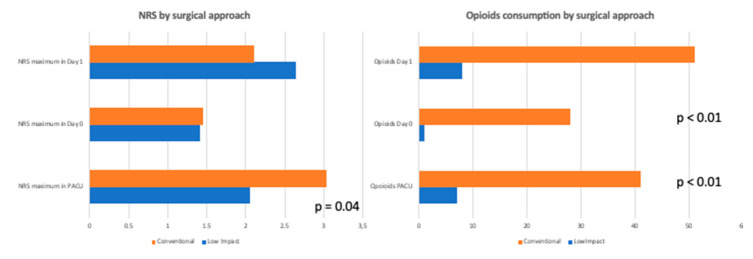
Postoperative opioid consumption and pain evaluation by surgical approach.

**Figure 2 jcm-11-02165-f002:**
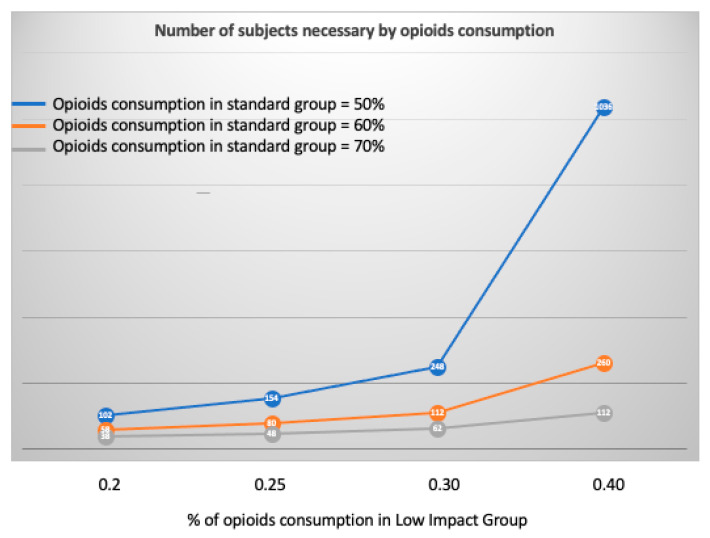
Theoretical number of patients necessary to demonstrate in a trial the benefit of low impact procedure on opioids consumption.

**Table 1 jcm-11-02165-t001:** Main patient characteristics.

	Conventional Laparoscopy N = 77 (%)	Low-Impact Laparoscopy N = 32 (%)	*p*-Value
Age, median (sd)	48.3 (13.0)	52.9 (12.5)	0.74
BMI ≥30	15 (19.5)	4 (12.5)	0.58
Menopausal	32 (41.6)	17 (53.1)	0.30
ASA physical status *			0.06
Class 1	29 (37.7)	20 (62.5)	
Class 2	42 (54.5)	11 (34.4)	
Class 3	6 (7.8)	1 (3.1)	
Comorbidity **	24 (31.2)	8 (25)	0.65
Increased bleeding risk	7 (9.1)	0	
Systemic disease	3 (3.9)	3 (9.4)	
Diabetes	5 (6.5)	0	
Hypertension	13 (16.9)	3 (9.4)	
Other	1 (1.3)	2 (6.3)	
Cesarean section			0.18
Category 1	14 (18.2)	2 (6.3)	
Category 2	9 (11.7)	6 (18.8)	
Category 3	1 (1.3)	0	
History of abdominal surgery			
Laparoscopy	22 (28.6)	10 (31.3)	0.9
Pfannenstiel laparotomy (except cesarean section)	10 (13.0)	1 (3.1)	0.17
Median laparotomy	7 (9.1)	2 (6.3)	1

* Missing data for 2 patients. ** Some patients had more than one comorbidity.

**Table 2 jcm-11-02165-t002:** Surgical factors.

Variables	Conventional N = 77 (%)	Low Impact N = 32 (%)	*p*-Value
Indication for surgery			0.01
Endometrial neoplasm			
Endometrial cancer	12 (15.6)	10 (31.3)	
Atypical endometrial hyperplasia	2 (2.6)	4 (12.5)	
Other conditions			
Cervical dysplasia	6 (7.8)	1 (3.1)	
Symptomatic Fibroma	37 (48.1)	10 (31.3)	
Adenomyosis	13 (16.9)	4 (12.5)	
Other	7 (9.1)	3 (9.4)	
Mean intraoperative pressure			<0.001
<10	18 (23.4)	28 (87.4)	
10–12	59 (76.6)	2 (6.3)	
>12	0	2 (6.3)	
Parietal adherences	20 (26.0)	6 (18.8)	0.57
Pelvic adherences	17 (22.1)	8 (25)	0.93
Deep pelvic endometriosis	1 (1.3)	3 (9.4)	0.08
Associated perioperative surgery *			
Uterosacral ligament resection	1 (1.3)	0	
Ureterolysis	1 (1.3)	5 (15.6)	<0.001
Parametrial resection	0	1 (3.1)	
Omentectomy, appendicectomy	0	2 (6.3)	
Median duration of anesthesia in minutes (Q1; Q3)	193 (167; 210.5)	174 (147; 206)	0.03
Median duration of surgery in minutes (Q1; Q3)	140 (118; 162)	129 (107; 151)	0.08
Median duration of pneumoperitoneum in minutes (Q1; Q3)	121 (98; 144)	105 (93; 129)	0.3
Median uterus size in mm (sd)	65 (22.5)	63 (21.3)	0.61
Median weight of the uterus in grammes (sd)	130 (108)	107 (136)	0.37

Unless otherwise specified, data are expressed in n (%). * Excluding adhesiolysis.

**Table 3 jcm-11-02165-t003:** Postoperative outcomes and pain.

	Conventional Laparoscopy N = 77 (%)	Low-Impact Laparoscopy N = 32 (%)	*p*-Value
Median length of stay in PACU in minutes (Q1; Q3)	136.5 (115; 161.2)	133 (114; 169)	0.74
Length of hospitalization (days)			<0.001
Outpatient	1 (1.3)	29 (90.6)	
24 h	39 (50.6)	2 (6.3)	
48 h	30 (39.0)	1 (3.1)	
>48 h	7 (9.1)	0	
Postoperative pain			
Maximum in PACU (mean; sd)	3.04; 2.46	2.06; 2.19	0.04
Maximum pain on Day 0 (mean; sd)	1.45; 1.84	1.42; 1.69	0.92
Maximum pain on Day 1 (mean; sd)	2.11; 1.95	2.64; 2.4	0.32
Ropivacaine port infusion	56 (72.7)	19 (59.4)	0.25
Analgesic Consumption			
Paracetamol			
PACU	22 (28.6)	9 (28.1)	0.96
Day 0	14 (18.2)	0	< 0.001
Day 1	17 (22.1)	2 (6.3)	0.055
Weak or strong opioid consumption anytime	49 (63.6)	10 (31.3)	*p* < 0.01
Weak opioids (Tramadol)			
PACU	6 (7.8)	0	<0.001
Day 0	8 (10.4)	0	<0.001
Day 1	23 (30.0)	4 (12.5)	0.09
Strong Opioids			
PACU	35 (45.5)	7 (21.9)	0.03
Day 0	20 (26.0)	1 (3.1)	<0.01
Day 1	28 (36.4)	4 (12.5)	0.02

**Table 4 jcm-11-02165-t004:** Analysis of the factors associated with ever consuming opioids in the postoperative period (PACU, Day 0 or Day 1).

Variables	Univariable Analysis	
	OR (95% CI)	*p*-Value
Age	1.0 (0.99–1.01)	0.76
BMI	1.01 (0.99–1.03)	0.20
Comorbidities	0.94 (0.77–1.16)	0.58
Indication for hysterectomy	1.04 (0.98–1.10)	0.20
Surgical approach	1.38 (1.13–1.69)	0.002
Mean intraoperative pneumoperitoneum pressure		
6–8	ref	
≥9	1.25 (1.03–1.51)	0.026
Mean intraoperative pressure (continuous)	1.043 (1.001–1.087)	0.046
Uterus size	1.00 (0.99–1.00)	0.58
Adherences	1.01 (0.83–1.23)	0.92
Duration of pneumoperitoneum	1.00 (0.99–1.00)	0.44
Ropivacaine port infiltration	1.11 (0.90–1.37)	0.33

## Data Availability

All relevant data are available within manuscript.

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
