# Peer review of "Total Hysterectomy by Low-Impact Laparoscopy to Decrease Opioids Consumption: A Prospective Cohort Study"

_jcm, 2022, doi:10.3390/jcm11082165_

Round 1
Reviewer 1 Report
This is a well-written paper with a study that provides evidence that Low-impact laparoscopy reduces postoperative pain and analgesic consumption compared to conventional laparoscopy. I have two questions below.
#1, Laparoscopic hysterectomy cause several types of pain including incisional pain, pneumoperitoneum, head-down position, the blood left in the abdomen, and dissection of the pelvic region.
Low-impact laparoscopy is a minimally invasive technique that combines low-pressure insufflation and microcoelioscopy, and It may reduce the pain especially caused by incisional pain and pneumoperitoneum. But I also think that you need to clarify the relation in the amount of bleeding and pain because It is well-known that the advantage of operating with high-pressure insufflation is the ease of obtaining intraoperative hemostasis.
I think it is important to show the amount of blood loss in surgical factors (Table 2).
#2, Please explain how did you select analgesics, and if you can clarify the rule or protocol, please write it in the method.
In the Low-impact laparoscopy group, How did the patient who went back home after surgery select the analgesics? And how did you educate them to choose pain medications at home?
You noted that patients in the Low Impact Group were prescribed paracetamol and tramadol at pre-operative appointments, but please clarify when and which patients were prescribed Strong Opioid.
In a conventional surgery group, how do medical staff chooses the analgesics? Did they have the protocol or guideline to select it?
Author Response
This is a well-written paper with a study that provides evidence that Low-impact laparoscopy reduces postoperative pain and analgesic consumption compared to conventional laparoscopy.
Thank you for this very positive comment
I have two questions below.
#1, Laparoscopic hysterectomy cause several types of pain including incisional pain, pneumoperitoneum, head-down position, the blood left in the abdomen, and dissection of the pelvic region.
Low-impact laparoscopy is a minimally invasive technique that combines low-pressure insufflation and microcoelioscopy, and It may reduce the pain especially caused by incisional pain and pneumoperitoneum. But I also think that you need to clarify the relation in the amount of bleeding and pain because It is well-known that the advantage of operating with high-pressure insufflation is the ease of obtaining intraoperative hemostasis.
I think it is important to show the amount of blood loss in surgical factors (Table 2).
The reviewer is right to underline the determinant role of the amount of bleeding on post-operative pain. This factor was not part of our table 2 since none of the patients experienced significant bleeding. This could be explained by two factors: 1) the experience of the surgeons that performed the surgeries 2) the relatively limited sample size.
We found this comment constructive and decided to specify these two elements in the results section and the limitation section of our revised manuscript.
#2, Please explain how did you select analgesics, and if you can clarify the rule or protocol, please write it in the method.
In the Low-impact laparoscopy group, How did the patient who went back home after surgery select the analgesics? And how did you educate them to choose pain medications at home?
Pain medications at home was function of pain intensity and medication required at hospital discharge. At home, patients were advised to always start with paracetamol when feeling pain and use weak opioids only when paracetamol was not efficient.
You noted that patients in the Low Impact Group were prescribed paracetamol and tramadol at pre-operative appointments, but please clarify when and which patients were prescribed Strong Opioid.
Pain medications were decided solely on the NRS. According to the NRS: Paracetamol for light pain (NRS < 4), Ibuprofen or weak opioids (Tramadol) for moderate pain (4 ≤ NRS < 7) and Morphine for intense pain (NRS ≥ 7).
This was specified in the revised version of our manuscript.
In a conventional surgery group, how do medical staff chooses the analgesics? Did they have the protocol or guideline to select it?
Medical staff, especially the nurses in the PACU, were completely blinded to the operative mode and pain medications were decided solely on NRS.
Reviewer 2 Report
The authors investigated in the present study the impact of low-impact laparoscopy on postoperative opioid consumption. The study is of scientific importance as it denotes the need to minimize intraabdominal pressure and trocar insertion site size. I believe that despite its significant selection bias (i wonder why did the authors opted not to perform an RCT which would significantly enhance their study) i believe that it deserves publication taking always into consideration its flaws.
I therefore suggest that the authors revise their manuscript by denoting this fact more consistently in their conclusion in which they should stress the need for RCTs and not cohort studies.
It would be interesting to mention if a pulmonary recruitment maneuver was used (Which is already known that it reduces significantly the need for postoperative opioids - especially in patients that do not undergo low-impact laparoscopy).
Author Response
The authors investigated in the present study the impact of low-impact laparoscopy on postoperative opioid consumption. The study is of scientific importance as it denotes the need to minimize intraabdominal pressure and trocar insertion site size. I believe that despite its significant selection bias (i wonder why did the authors opted not to perform an RCT which would significantly enhance their study) i believe that it deserves publication taking always into consideration its flaws.
Thank you for this positive comment. The reviewer is right: the best way to definitely assess the benefit of low impact surgery would have been to conduct a RCT. However, since little data exist on the value of low impact procedures, we aimed to perform this first study by using “daily” practice in our center to better plan an RCT and to define the appropriate number of subjects to include.
I therefore suggest that the authors revise their manuscript by denoting this fact more consistently in their conclusion in which they should stress the need for RCTs and not cohort studies.
This part was emphasized in the revised version of our manuscript and especially in the conclusion.
It would be interesting to mention if a pulmonary recruitment maneuver was used (Which is already known that it reduces significantly the need for postoperative opioids - especially in patients that do not undergo low-impact laparoscopy).
Indeed, this information would be of interest but was not systematically collected during the study. This will be part of the variables studied in the RCT that we are currently initiating on this matter. This comment has enriched the limitation section of our work.